# Simulation of Layer Thickness Measurement in Thin Multi-Layered Material by Variable-Focus Laser Ultrasonic Testing

**DOI:** 10.3390/s23020694

**Published:** 2023-01-07

**Authors:** Jinxing Qiu, Zhengying Li, Cuixiang Pei, Guoqiang Luo

**Affiliations:** 1School of Information Engineering, Wuhan University of Technology, Wuhan 430070, China; 2Shaanxi ERC of NDT and Structural Integrity Evaluation, State Key Laboratory for Strength and Vibration of Mechanical Structures, Xi’an Jiaotong University, Xi’an 710049, China; 3State Key Laboratory of Advanced Technology for Materials Synthesis and Processing, Wuhan University of Technology, Wuhan 430070, China

**Keywords:** laser ultrasonic testing, non-destructive testing, ring-shaped laser, thickness measurement, multi-layered material

## Abstract

Thin multi-layered materials are widely used in key structures of many high technology industries. To ensure the quality and safety of structures, layer thickness measurement by non-destructive testing (NDT) techniques is essential. In this paper, a novel approach for the measurement of each layer’s thickness in thin multi-layered material is proposed by using ring-shaped laser generated focused ultrasonic bulk waves. The proposed method uses a ring-shaped laser with a variable radius to generate shear waves with variable focus inside the structure. By analyzing the signal characteristics at the ring center when the laser radius varies from zero to maximum, the direct measurement of layer thickness can be realized, considering that only when the focal depth and the layer thickness satisfy the specific relationship, the reflected shear waves converge and form a peak at the ring center. This straightforward approach can increase the pulse-echo SNR and prevent the processing of aliasing signals, and therefore provides higher efficiency and accuracy for the layer thickness measurement. In order to investigate the feasibility of this method, finite element simulations were conducted to simulate the ring-shaped laser generated ultrasonic waves in multi-layered structure in detail. Following the principle of the proposed method, the layer thickness of a bi-layer and 3-layer structure were respectively measured using simulation data. The results confirm that the proposed method can accurately and efficiently measure the layer thickness of thin multi-layered material.

## 1. Introduction

Thin multi-layered materials such as functional gradient materials, polymer laminates, and thermal barrier coatings are frequently employed in major aerospace engineering, energy, and other high-technology industrial structures [1,2,3]. The thickness of each layer must be measured using non-destructive testing (NDT) methods to assure the quality and safety of such structures because variations in layer thicknesses might reduce the mechanical qualities of the structure, or possibly result in its failure.

For measuring material thickness, several NDT techniques have been reported until now [4,5]. One of the most widely used NDT techniques among them is ultrasonic testing (UT), which has been used extensively for thickness measuring, material performance testing, and defect inspection [6]. However, it is challenging to use standard UT, such as the pulse-echo based approach, to measure the layer thickness of thin multi-layered material. The reasons can be listed as follows: (1) The overlap of ultrasonic echoes from interlayers is severe because layer thickness is narrow (often in the micrometer range). (2) Echoes from deep levels are greatly diminished as a result of the dispersion and reflection of numerous contacts. The severe signal aliasing and poor SNR of interface pulse echoes make it difficult to extract and analyze the time of flight information of ultrasonic signals. Current research focuses on the separation of overlapped pulse echoes using signal processing [7,8,9] and parameter optimization using the inversion algorithm [10,11,12,13] to address the aforementioned issues. However, the method of indirect thickness evaluation is vulnerable to a difficult inversion process or subpar evaluation accuracy.

Laser ultrasonic testing (LUT), a recent addition to the UT family, has demonstrated its capacity to produce ultra-high frequency and broadband ultrasonic waves with a non-contact characteristic. In several earlier studies, LUT has been researched and used for inspecting surface or near-surface defects. In addition, several reports [14,15] have appeared where thickness measurements were made utilizing longitudinal waves produced by lasers. Related investigations that use the pulse-echo approach, nevertheless, have found that signal aliasing issues are still present. Additionally, the thermoelastic regime generates bulk waves with weak signals. This effect significantly reduces the pulse echo’s SNR in deep layers and restricts its use in the assessment of thickness. Research has been conducted on the modulation of pulse laser patterns, such as the ring-shaped laser ultrasonic testing (RLUT) approach, to increase the signal amplitude without damaging the material surface by ablation. In contrast to surface waves produced by a line source, P. Cielo et al. [16] have reported a 20-fold increase in signal level using a ring-shaped pulsed laser to generate convergent surface waves. To find cracks on the bottom surface, X. Wang et al. originally employed ring-shaped laser-generated ultrasound [17]. The theoretical model and finite element simulation of an acoustic wave produced by a ring laser were developed by J. Guan et al. [18]. The benefits of ring focused laser ultrasound in fracture identification over non-focused laser ultrasound were further enhanced by S. Dixon [19]. The superposition of generated bulk waves is made possible at a particular focus point by the use of a ring-shaped laser. SNR and spatial resolution can be significantly increased as a result. The research reports mentioned above, however, only focused on the inspection of surface or interior flaws and cannot be used to assess the thickness of thin material, particularly in multi-layered structures.

In this study, a novel variable-focus ring-shaped laser ultrasonic testing (VRLUT) technique is proposed for measuring the layer thickness of thin layered material. The ultrasonic waves with varied focus are generated inside the structure by repeatedly exciting ring-shaped lasers of various diameters on the material surface. Our research found that only when the focal depth and the layer thickness satisfy the specific relationship do the reflection waves converge at the ring center. This phenomenon can be witnessed from A-scan signals, and subsequently allows the analysis of the quantitative relationship between focal depth and layer thickness. By matching the focal depth and layer interface locations, direct thickness measurement is then possible, avoiding issues with traditional approaches, such as complex parameter inversion and the separation of echo signals with severe aliasing. Additionally, the employment of deep-focused waves can successfully raise the signal-to-noise ratio (SNR) of echoes from deep layers, which have a high degree of attenuation and dispersion.

This paper is organized as follows. The principle of the proposed method is introduced in Section 2. Numerical simulations using the finite element method (FEM) to examine the viability of this approach is introduced in Section 3. Measurement results for various thin stacked metal plates by using simulation data are given in Section 4. Finally, the conclusion is presented in Section 5.

## 2. Principle of VRLUT-Based Layer Thickness Measurement Method

### 2.1. Focus Pattern of Laser-Generated Bulk Waves

There is a directivity pattern in the ultrasonic bulk waves produced by lasers. The amplitude of longitudinal and shear waves generated by thermoelastic expansion is a function of angle, and is defined in Equations (1) and (2) [20].
(1)$$u_{L}\propto \frac{\sin\theta \sin2\theta \sqrt{\kappa^{2}-\sin^{2}\theta }}{{\left(\kappa^{2}-2\sin^{2}\theta \right)}^{2}+4\sin^{2}\theta \cos\theta \sqrt{\kappa^{2}-\sin^{2}\theta }}$$
(2)$$u_{s}\propto \frac{\sin2\theta \left(1-2\sin^{2}\theta \right)}{{\left(1-2\sin^{2}\theta \right)}^{2}+4\sin^{2}\theta \cos\theta \sqrt{\kappa^{-2}-\sin^{2}\theta }}$$
where

$u_{L}$: Amplitude of longitudinal wave,

$u_{L}$: Amplitude of shear wave,

θ: Angle to the surface normal,

$\kappa =C_{L}/C_{S}$: Ratio of the longitudinal wave velocity to the shear wave velocity.

Figure 1 shows the directivity patterns of the laser-generated ultrasonic waves in an iron plate with *C_L_*/*C_S_* = 1.86. While the amplitude of the shear wave rapidly decreases in other directions, it exhibits a dramatic peak at around 34° normal to the surface. In contrast, the longitudinal wave’s amplitude exhibits a peak at around 69° normal to the surface, but does not rapidly decelerate in other directions.

If the ring laser source is used as a point source assessment, each point source will produce a shear wave and a directive longitudinal wave that overlap at the angle of maximum amplitude. Then, at a particular focal depth below the center of the laser ring, a focal area is generated. The focal area, particularly for shear waves, is so narrow that it can be regarded as a point. As a result, this study focused on the application of shear waves and their primary characteristics.

### 2.2. Layer Thickness Measurement Method

The designed system and principle for thickness measurement using RLUT are shown in Figure 2. The initial point-like laser beam is generated by a Q-switch solid-state pulsed laser with duration ≤ 8 ns, wavelength of 1064 ns, and maximum pulse energy of 80 mJ. A beam expander initially enlarges the initial point while measuring. After that, the enlarged beam is sequentially passed via a diffractive axicon (DA) and a convex lens (CL). The focused laser source is then created on the specimen surface in the shape of a ring. To change the distance to the specimen, the CL and DA are mounted on two micro-displacement scanning stages with stepping accuracy of 6.5 μm. The ring radius can be modified by changing the DA while keeping the ring width constant because the ring width only depends on the distance between the convex lens and the specimen. Therefore, by gradually increasing the distance between DA and the specimen, it is possible to irradiate the specimen surface with a ring-shaped laser whose radius continually varies from zero to maximum. Through the use of a laser interferometer and a dichroic mirror (DM), the surface displacement brought about by the generated ultrasonic waves is detected. The laser interferometer can be a single-point heterodyne type with bandwidth of at least 500 MHz. A high-speed oscilloscope with bandwidth of 30 GHz and sampling rate of 80 GS/s is used to record and later process the ultrasonic signals that the interferometer observed. The testing should be conducted at room temperature and indoor environment to ensure the accuracy of optical measurement system.

As depicted in Figure 3a, while performing the measurement, the ring-shaped laser, with radius varying from zero to maximum, is generated on the specimen surface to conduct the R-scan procedure. Supposing the reflection angle of reflected bulk waves is absolutely the same as incident angle of initial waves, the superposition of reflected shear waves will be observed at the ring center if the focal depth *d*, angle of maximum shear amplitude *θ*, and the ring radius *R* follow:2*d* = *R*tan*θ*,(3)

As shown in Figure 3b, the reflection waves will converge in the center of the laser ring as illustrated by the blue lines, and a peak will be visible in the time-domain waveform of the center while the ring radius *R* fits the layer thickness *h*, which means *h* equaling *d*. In other cases, peaks will occur at the ring around the center, but not exactly at the center, as illustrated by the red lines.

Figure 4 displays the typical RLUT signal at the ring center when *R* matching *h* and *R* unmatching *h*. Due to its higher velocity compare to other waves, the surface longitudinal wave (sL) in both situations arrives at the ring center first. Due to the very modest shear wave component in this direction (as shown in Figure 1), the surface shear wave (sS) with low amplitude then emerges. The Rayleigh wave has a substantially bigger amplitude than other waves because of its comparatively high initial amplitude and center overlap. Different phenomena may then be seen: when *R* matches *h*, the superposition of reflection waves (2S) occurs at the center, and when *R* unmatches *h*, the amplitude is significantly lower than in the former situation. This allows one to determine which *R* best matches the *h* after evaluating the signals produced by various radius ring-shaped lasers and contrasting the signals at the center. Then the thickness of the first layer can be calculated by Equation (3). It should be noted that only shear waves are employed in this method since they are straightforward to identify, have a high signal-to-noise ratio, have a smaller maximum amplitude angle, and have a more evident superposition effect than longitudinal waves. However, for a particular ring radius, not only the shear echo but also the longitudinal echo and reflected mode-converted waves may overlap at the center and give a deceptive indication of the matching *R*. Therefore, the relationship between the flight time and propagation distance is utilized to further verify the calculated *h* using:
(4)$$c_{s1}t_{s1}=2\sqrt{{\left(R/2\right)}^{2}+h_{1}^{2}},$$
where *C_s__i_* and *t_s__i_* are the shear wave velocity and flight time in *i*th layer, respectively.

Only when the calculated *h* simultaneously satisfies Equations (3) and (4), the *h* is regarded as a true value. For the second layer, as shown in Figure 3c, Equation (3) can be written as:(5)$$R=2\left(h_{1}\tan\theta_{1}+h_{2}\tan\theta_{2}\right),$$

According to Snell’s law:(6)$$\frac{\sin\theta_{1}}{\sin\theta_{2}}=\frac{c_{s1}}{c_{s2}},$$

The thickness of the second layer can be estimated once the first layer’s thickness and the shear wave velocities in all layers are known. Following the flight time of superposition shear waves at the center point, the predicted thickness should also be:(7)$$\frac{2h_{1}}{\cos\theta_{1}}+c_{s2}\left(t_{s2}-\frac{2h_{1}}{c_{s1}\cos\theta_{1}}\right)=2\left(\frac{h_{1}}{\cos\theta_{1}}+\sqrt{{\left(\frac{R}{2}-h_{1}\tan\theta_{1}\right)}^{2}+h_{2}^{2}}\right),$$

The thickness measurement of a bi-layer material is then achieved. In the same way, it is possible to determine each layer’s thickness layer by layer in a multi-layered structure with *n* layers by:(8)$$R=2\sum_{i=1}^{n}h_{i}\tan\theta_{i},$$
where *h_i_* and *θ_i_* are respectively the thickness and angle of maximum shear amplitude in *i*th layer.

## 3. Simulation of Thickness Measurement by VRLUT

### 3.1. Simulation Model and Main Parameters

The proposed thickness measurement technique was validated by the analysis of ultrasonic waves using the finite element method (FEM). A commercial software COMSOL was used to conduct the FEM simulation. Two numerical models were developed, as shown in Figure 5, to depict the thickness measurement process and examine the precision of the proposed method. A cylindrical coordinate system is also introduced in our model to reduce computation costs. As a result, the issue can be thought of in the axial symmetric plane of two dimensions. The simulation models contain a W-Cu bi-layer plate and an Al-Fe-Cu three-layer plate. On the top surface of the first layer in both models, a ring-shaped laser is irradiated with energy below the sample damage threshold. The ring radius and half of the irradiation width are respectively denoted by *R* and *w*. The thickness of each layer of the n-layer material is denoted by *h_i_* (*i* = 1, 2, …… *n*) from top to bottom. The spatial distribution *f*(*r*) and temporal distribution *g*(*t*) of the laser source in the simulation model can be expressed as:(9)$$f(r)=\exp(-{\left(r-R\right)}^{2}/w^{2}),$$
(10)$$g(t)=8t^{3}/t_{0}{}^{3}\exp(-2t^{2}/t_{0}{}^{2}),$$
where *t*_0_ is the rise time of the laser pulse.

According to some reports, the laser energy, ring radius, and width can all have an impact on the laser-generated ultrasonic waves [21,22]. To obtain a significant signal amplitude, the laser energy should be set as high as feasible while yet staying below the thermoelastic mode threshold. The accuracy of the measurement can be increased by employing a thin width ring laser since the frequency of ultrasound has an inverse relationship with ring width. However, the diffraction limit of the optical components places a limit on the ring width, and a smaller laser width would require finer meshing, increasing the cost of the calculation. To balance the calculation cost and the accuracy of the numerical results, the laser pulse width and laser ring width are assumed to be 1 ns and 2 μm respectively. The laser’s peak power density was set at 0.012 W/m^−2^. During the R-scan procedure, the radius of the laser ranges from 0 to 500 μm with a step of 2 μm. The element size of the grid was set as 0.1 μm in the vicinity of the laser-affected area, while the element size was set at 0.5 μm outside the heat-affected zone. Details of material parameters used in the simulation are listed in Table 1. All parameters are from COMSOL’s built-in materials.

### 3.2. Calibration of Wave Velocity and Focal Angle

Before the application for thickness assessment, simulation is used to confirm the focal properties of the shear wave produced by the ring-shaped laser. Moreover, the maximum shear amplitude angle might not match the theoretical value because of the attenuation of ultrasonic waves and the fact that the focus region is not a precise point. The velocity of the shear wave is also unknown, according to Table 1. Therefore, it is crucial to calibrate the maximum shear amplitude angle and wave velocity before the measurement procedure. In order to calibrate and determine the wave velocity and focal angle, simulations were firstly conducted by using a one-layer model. In this model, a ring-shaped laser with radius of 50 μm and width of 2 μm is used to irradiate the top surface of a copper plate with thickness of 100 μm. The 3D view of the simulated ultrasonic field inside the plate at different times is shown in Figure 6. The bulk waves have not yet overlapped at time t = 10 ns. Then at t = 15 ns, the longitudinal waves began to overlap; however, because the focus effect was not obvious, the amplitude of the waves in the overlapped area was relatively low. At t = 30 ns, the shear waves overlap, and an obvious focus point is observed. Finally, at t = 50 ns, the focus point vanished after the shear waves split. The above results were helpful in understanding the focus pattern of bulk waves, and it could be clearly observed that shear waves perform better in focus.

The calibration of shear wave velocity and focal angle can be then accomplished by using above simulation results. From the geometrical relationship as shown in Figure 6c, the focal angle *θ* can be obtained by:(11)$$\tan\theta =f_{d}/R,$$
where *R* is the radius of laser ring and *f_d_* is the distance between laser ring center and focal point. The shear wave velocity *v*_s_ is obtained according to the flight time *t*:(12)$$v_{s}=\sqrt{f_{d}{}^{2}+R^{2}}/t,$$

As the focal point is thought to be located at the coordinate of the maximum peak value, the wave amplitudes along the Z-axis were then extracted to find out the coordinate of focal point. The extracted amplitudes of the shear wave along the Z-axis from 38 ns to 42 ns are shown in Figure 7. The shear wave’s amplitudes exhibit a unimodal distribution, as expected, and the maximum peak-to-peak amplitude is attained at a position close to Z = 36.8 μm with flight time t = 40 ns. The velocity of shear wave *v*_s_ = 2040 m/s and focal angle *θ* = 38.0° can then be calculated according to Equations (11) and (12), which is larger than the angle theoretically analyzed by Equation (2). The reason for the error is that the propagation distance of ultrasonic wave in this direction is smaller than that in the theoretical one, which means the attenuation is simultaneously smaller. These calibrated values served as the ideal values in this study. Similarly, the velocity of the shear wave and the focal angle in the aluminum plate is calculated as *v*_s_ = 3014 m/s and the focal angle *θ* = 36.4°.

## 4. Measurement Results

By measuring the thickness of multilayered materials, the performance of the suggested method is examined to validate it. In Section 3.1, the simulation model and computation setup have been shown. The bi-layer model’s typical simulation results for the volumetric ultrasonic displacement field at various times when exposed to a 400 μm radius ring laser are shown in Figure 8. While there is no focus to be seen in Figure 8a,c, Figure 8b shows a large amplitude of displacement in the center of the ring that indicates a good convergence of shear echoes. The time domain signals of the ring center are extracted as illustrated in Figure 9a,b. Only when R varies near 190 μm and 410 μm, which correspond to the thickness of the first layer and second layer, respectively, can the peak value behind the Rayleigh wave, indicating the focal shear echoes, be seen. Then the amplitude of peak value Δ*d* is furtherly compared, as shown in Figure 10. Finally, the optimum matching radius for the two-layer is selected as 190 μm and 412 μm. The thickness is determined to be 121.59 μm and 80.54 μm, respectively, via Equations (3) and (5). Additionally, the computed thickness values are verified by flight duration. The predicted flight times of overlapped shear echoes are estimated by Equations (4) and (6) to be around 151 and 255 ns, which is quite similar to the values seen in Figure 10. The simulation results for the 3-layer model presented in Figure 11 demonstrate a similar tendency. The optimum matching radii for the three layers were selected as 74 μm, 152 μm, and 178 μm, and the thicknesses were calculated as 50.19 μm, 50.30 μm, and 30.36 μm, respectively. The thickness measurement results of each layer are listed in Table 2. The aforementioned results demonstrate that the suggested method can accurately determine the thickness of every layer with an error of less than 1.5%. Additionally, each measurement step takes 200 ns for signal acquisition and 0.2 s for automatically adjusting R, resulting in a total measurement time of less than 1 min for the 250 steps when R ranges from 0 to 500 μm.

It should be noted that the proposed method and all results demonstrated above are valid only when following conditions are met: (1) The material in each layer is isotropic and homogeneous. In this case, the propagation of shear wave in a layer can be regarded as along one direction. (2) The total thickness of tested specimen is much smaller than other dimensions, so that the wave reflection of the specimen edge does not produce a large influence on the signal analysis. Due care should also be taken to ensure the reproducibility of the obtained values: (1) The distribution of laser power density along the ring direction should be uniform. (2) The surface of the specimen should be smooth and clean. In this way, we can ensure that, in each measurement, the amplitude of focal shear echoes at the ring center is able to keep constant and reach maximum when R matches the thickness. In practical applications, an optical homogenizer may be introduced to improve the uniformity of the laser beam when the laser quality is poor. For the tested specimen of which the surface condition is undesirable, proper surface treatment such as polish operation or painting may be desired.

## 5. Conclusions

In this study, a variable-focus ring-shaped laser ultrasonic testing (VRLUT) technique is proposed for the measurement of each layer’s thickness in thin multi-layered material. The matching feature between layer thickness and focus position is retrieved by examining the signal characteristics at various relative positions between the focus point and the multilayer interface. As a result, it is possible to directly quantify layer thickness using the matching relationship between the radius of laser ring and the thickness of the layer combining with the flight time of focused echoes. The proposed method may deliver a high SNR of measuring echoes while also avoiding the complicated data processing process, which is able to effectively measure the layer thickness of thin multi-layered structures in practice with higher accuracy and speed compared to current approaches. Furthermore, as a fully non-contact testing method, the proposed technique has the potential to be applied for the on-line measurement of material thickness during manufacturing.

The proposed technique is validated with use of simulated signals to assess the layer thickness of a bi-layer and 3-layer structure. The results show that the developed method is well adapted to measure multiple layer thicknesses that are even less than 50 μm. For both the bi-layer and 3-layer structure, their layer thickness was obtained with relative error of less than 1.5%, which proved the validity of proposed method and corresponding algorithm. The above findings suggest that the VRLUT-based approach can offer a very efficient and precise NDT technique for assessing multiple layer thicknesses in thin multi-layered material.

Furthermore, the measurement system will be established, and experimental research needs to be conducted to examine the realism and applicability of these novel methodologies for measuring layer thickness.

## Figures and Tables

**Figure 1 sensors-23-00694-f001:**
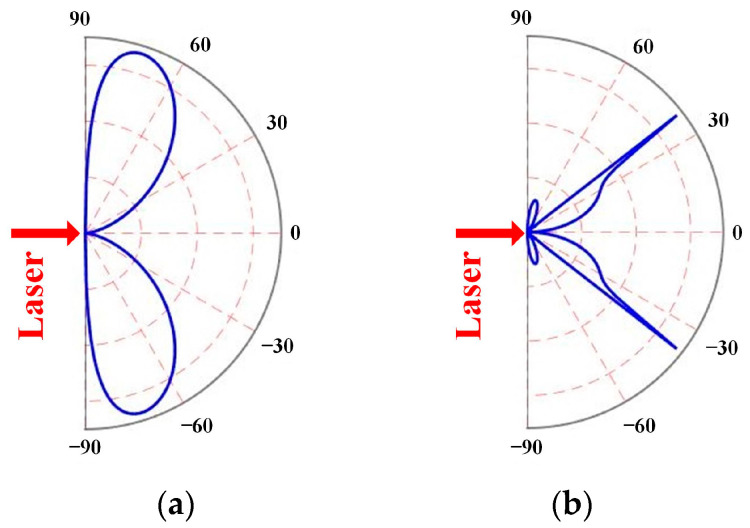
Directivity patterns of laser-generated ultrasonic waves in iron: (**a**) longitudinal wave and (**b**) shear wave in thermoelastic regime.

**Figure 2 sensors-23-00694-f002:**
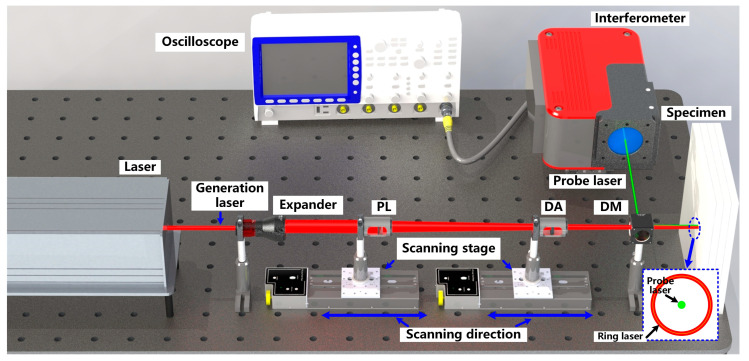
Schematic diagram of RLUT system for thickness measurement.

**Figure 3 sensors-23-00694-f003:**
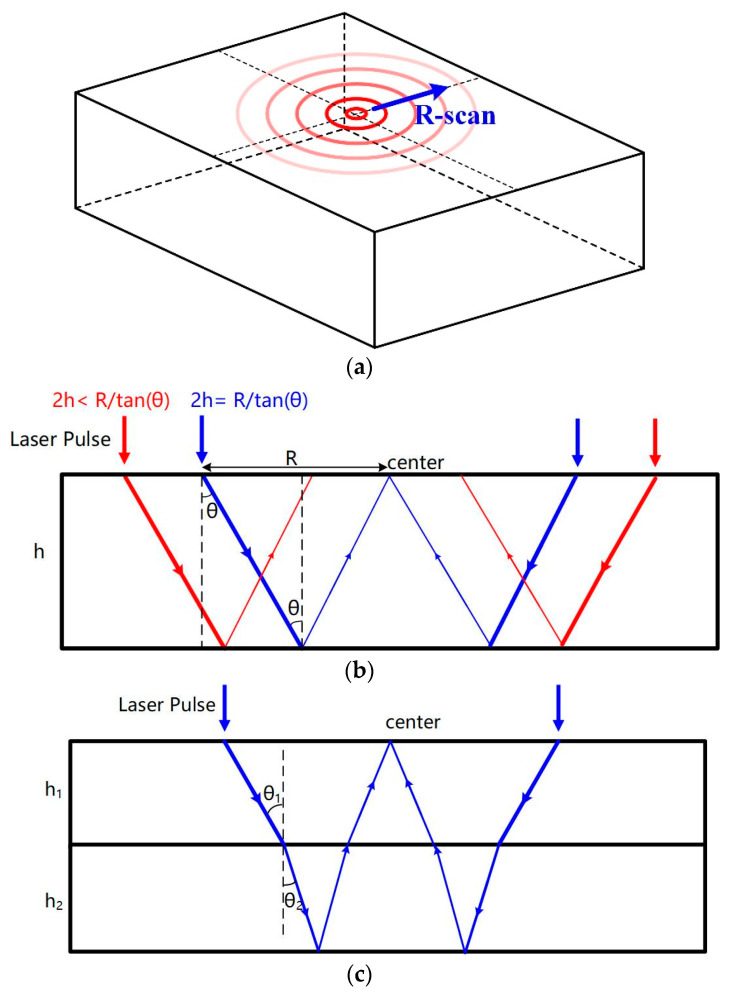
Schematic diagram of (**a**) ring-shaped laser irradiation and cross-sectional view of wave propagation in (**b**) a single-layer material and (**c**) a bi-layer material.

**Figure 4 sensors-23-00694-f004:**
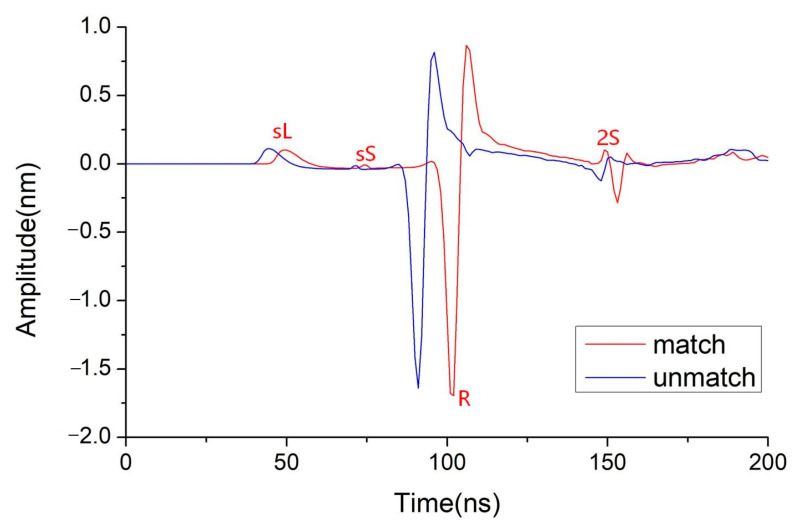
Typical RLUT signal: the ring radius *R* matches the layer thickness *h* vs. *R* unmatches *h*.

**Figure 5 sensors-23-00694-f005:**
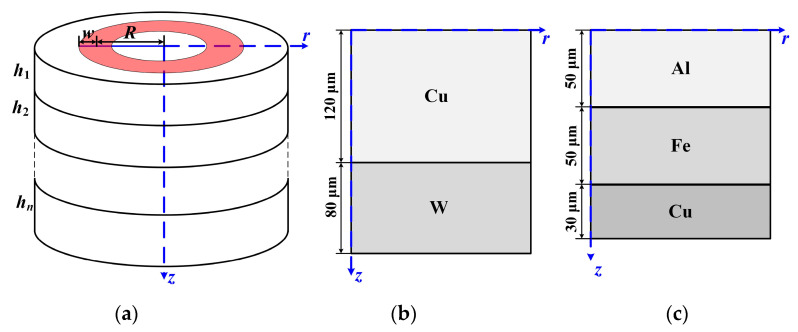
Schematic diagram of the simulation model: (**a**) 3D view; (**b**) half cross-section view of the Cu-W bi-layer plate; (**c**) half cross-section view of the Al-Fe-Cu three-layer plate. The red zone represents the laser irradiated area.

**Figure 6 sensors-23-00694-f006:**
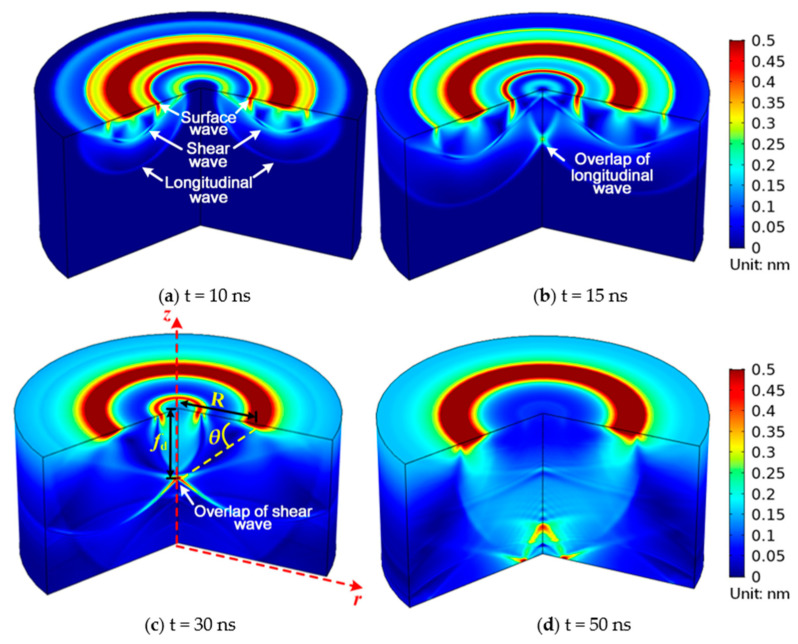
Cut plane view of the volumetric ultrasonic displacement field at different times: (**a**) before bulk waves overlap; (**b**) longitudinal waves overlap; (**c**) shear waves overlap; (**d**) after bulk waves overlap.

**Figure 7 sensors-23-00694-f007:**
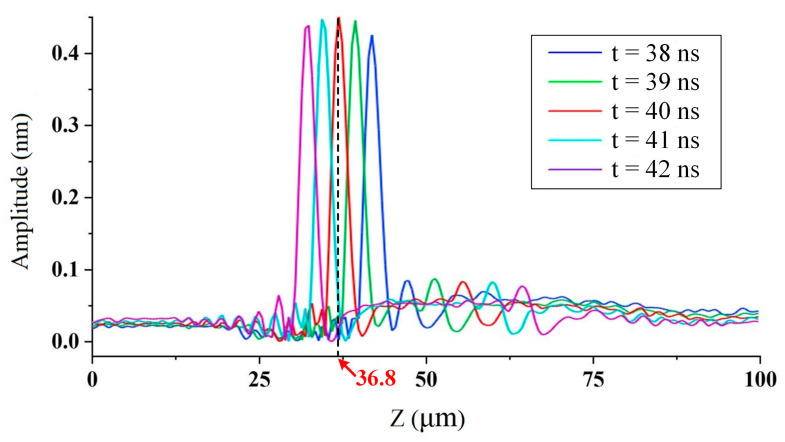
Amplitudes of the shear wave along the central axis near the focus time.

**Figure 8 sensors-23-00694-f008:**
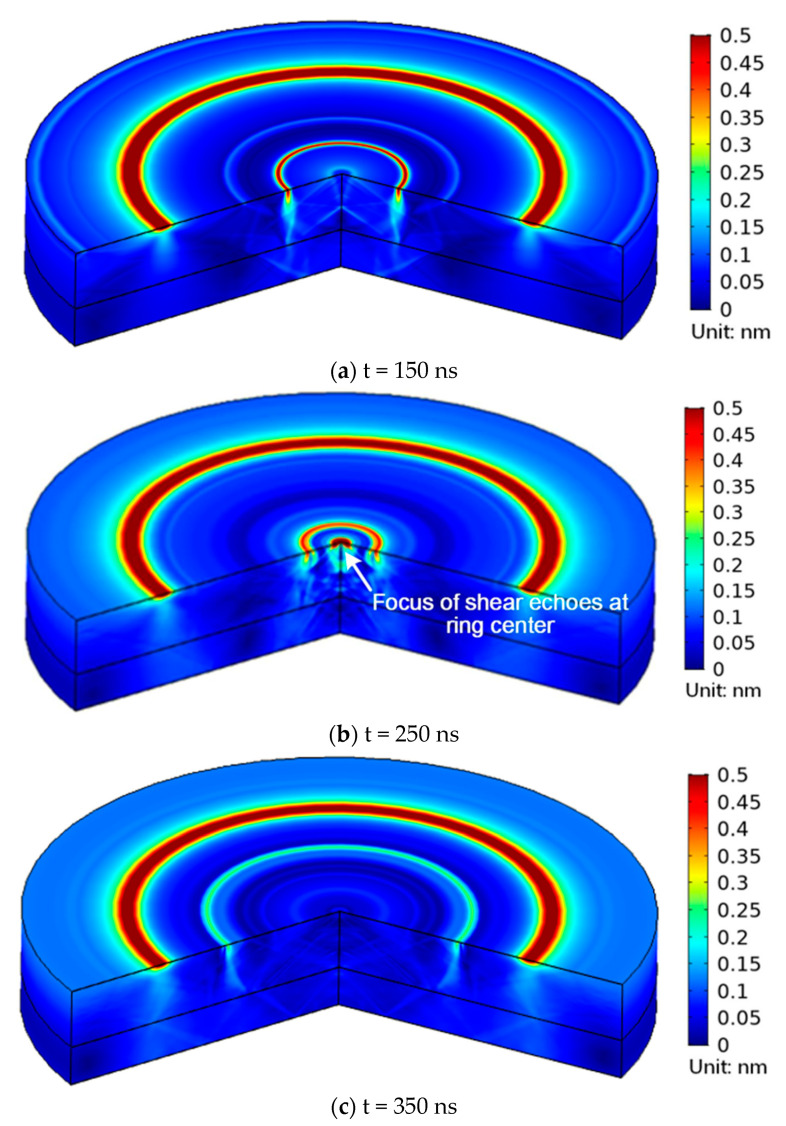
Volumetric ultrasonic displacement field at different times for *R* = 410 μm.

**Figure 9 sensors-23-00694-f009:**
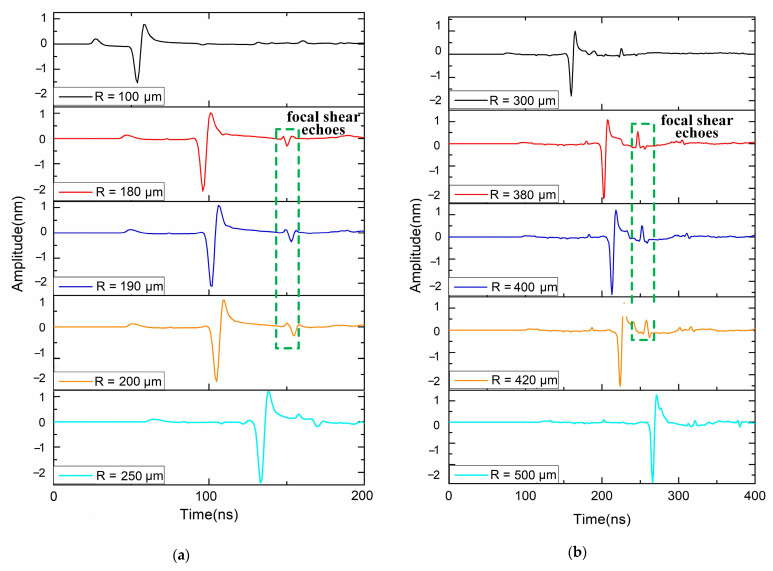
Typical simulation results of displacement at the ring center for the thickness measurement of (**a**) 1st layer and (**b**) 2nd layer.

**Figure 10 sensors-23-00694-f010:**
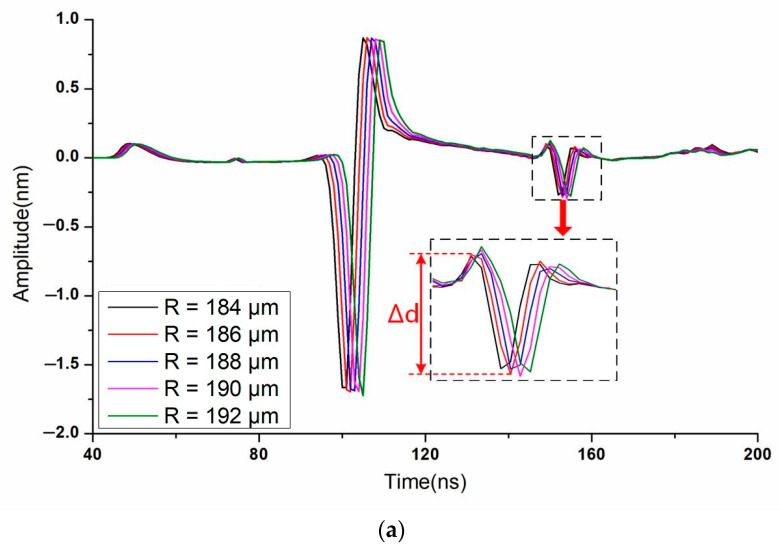
Simulation results of displacement at the ring center and comparison of peak-to-peak amplitudes of the focal shear echoes when *R* varies near (**a**) 190 μm and (**b**) 410 μm.

**Figure 11 sensors-23-00694-f011:**
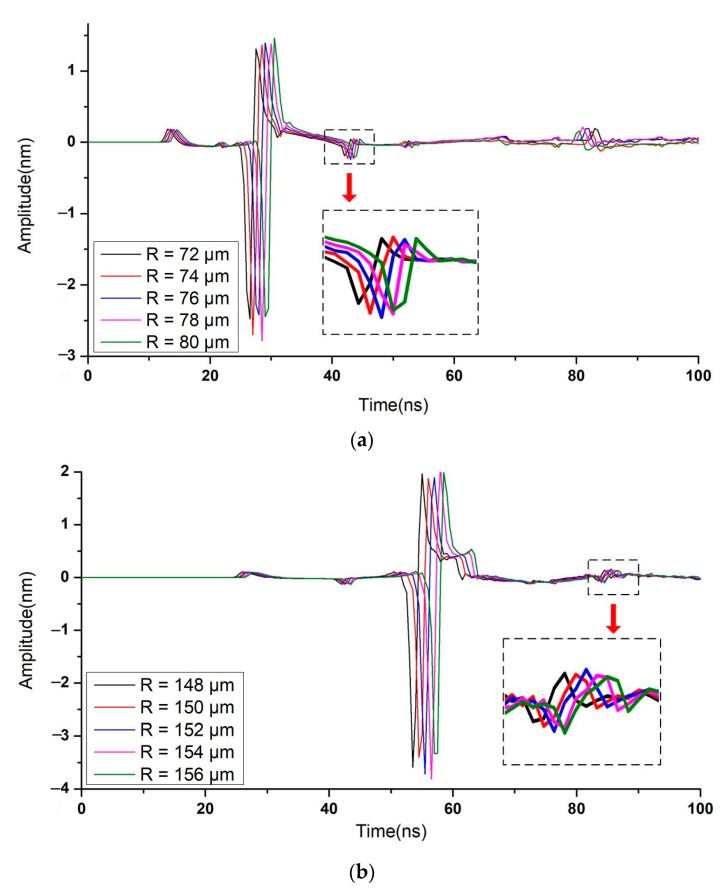
Simulation results of displacement at the ring center and comparison of peak-to-peak amplitudes of the focal shear echoes when *R* varies near (**a**) 80 μm, (**b**) 150 μm, and (**c**) 180 μm.

**Table 1 sensors-23-00694-t001:** Material parameters used in simulation cases.

Layer	Thermal Conductive Coefficient λ (W·m^−1^·K)	Thermal Capacity c (J·kg^−1^·K^−1^)	Density *ρ* (kg·m^3^)	Young’s Modulus *E* (GPa)	Poisson’s Ratio σ	Thermal Expansion Coefficient *α* (K^−1^)
Copper	400.0	385	8960	110	0.35	1.70 × 10^−5^
Tungsten	175.0	132	17800	360	0.28	0.45 × 10^−5^
Aluminum	238.0	900	2700	70	0.33	1.70 × 10^−5^
Iron	76.2	440	7870	200	0.29	1.22 × 10^−5^

**Table 2 sensors-23-00694-t002:** Layer thickness measurement results.

Layer	Measured Thickness	Real Thickness	Relative Error
Copper (120 μm)	121.59 μm	120.00 μm	1.33%
Tungsten	80.54 μm	80.00 μm	0.68%
Aluminum	50.19 μm	50.00 μm	0.38%
Iron	50.30 μm	50.00 μm	0.60%
Copper (30 μm)	30.36 μm	30.00 μm	1.20%

## Data Availability

The data presented in this study are available on request from the corresponding author.

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
