# Peer review of "Simulation of Layer Thickness Measurement in Thin Multi-Layered Material by Variable-Focus Laser Ultrasonic Testing"

_sensors, 2023, doi:10.3390/s23020694_

Round 1
Reviewer 1 Report
In this paper authors present a study conducted to simulate the thickness measurement of thin layered structures using ring-shaped laser to generate ultrasonic waves. The simulation is done via finite elements using comsol. Although the approximation is interesting and the analytical study is clear and sound, providing equations to solve the problem, authors merely present a simulation using conventional FEM methods. It is a pitty they have not made real experiments and compare results with simulations, once they even include an experimental set-up. If there is no experimental data, the word “simulation” should be in the title, otherwise it may lead to the conclusion that there is an actual system working and providing good results, which is not the case, according to the paper, or at least it is not mentioned in the paper. The work is good for a simulation and the statement of an idea, but for publication it should be accompanied with experimental data.
Reviewer 2 Report
Following the review of the paper "Thickness measurement of thin multi-layered material using ring-shaped laser generated variable-focus ultrasonic waves", we can conclude that:
1. In my opinion, the abstract of the paper is far too synthetic. It is necessary to add in the abstract more relevant aspects that is presented in the present version of the paper;
2. Regarding of the measurements and experimentations: it is necessary for the authors to indicate in the paper, all type of the equipment that is used in order to obtain the experimental values. In addition, it is necessary to clarify in the paper all the measurement conditions for the experiments performed;
3. Regarding the experimental results it is necessary for the authors to improve the discussion regarding the calibration of the measurement system that is indicated in Figure 2. „Schematic diagram of RLUT system for thickness measurement”. In this regard, it can be said that Figure 2 is incomplete because the wiring shown in Figure 2 is incorrect; the oscilloscope is no connected (the electrical wiring is not shown in the Figure 2). Additional figures in this respect would be desirable. The only explanation in this sense is, as the authors themselves say, “The shear wave's amplitudes exhibit a unimodal distribution, as expected, and the maximum peak to peak amplitude is attained at a position close to 36.8μm with a time constant of 40 ns. The velocity of shear wave vs = 2040 m/s and focal angle θ = 38.0°can then be calculated, which is larger than the angle theoretically analyzed by Equation (2). These calibrated values served as the ideal values in this study”;
4. A parallel made between the results obtained experimentally and those obtained by numerical simulation would be desirable. It is necessary for the authors to clarify this important aspect;
5. In general, it is elegant that a paragraph does not end with a figure or table, but with a comment. In this regard, is necessary to add a comment after Figure 1 and Figure 7, Table 1 and Table 2. It is also elegant that a paragraph does not end with a relation i.e. (8), but with a comment. In this regard, is necessary to add a comment after relation (8);
6. It would be interested to authors highlight all limitations of their method developed in this paper;
7. It is necessary for the authors to improve in the paper the discussion regarding the reproducibility of the obtained values;
8. The conclusions of the paper must be comprehensive and well-organized information; the paper contains many valuable results that need to be highlighted in the conclusions. The conclusions of the paper must contain the possible implications of these study in future practical developments. What are the prospects for capitalizing on this research? It is necessary to add and these aspects to the conclusions.
Round 2
Reviewer 1 Report
Authors have followed my suggestions.
Reviewer 2 Report
Following the review of the paper "Simulation of layer thickness measurement in thin multi-layered material by variable-focus laser ultrasonic testing", in the present revised form, it can be said that the paper have potential applications and the proposed new methodology is ready to be implemented in practice. The authors of the paper should publish a paper in the near future with results obtained from practical experiments, in which a parallel made between the results obtained experimentally and those obtained by numerical simulation.
- I have looked very carefully on the answers of the authors, regarding on all recommendations. In addition, I have looked carefully of the entire paper, in the present revised form. I think that the authors solved correctly and thoroughly all of the recommendations, with the exception of recommendation No. 4;
- Since the paper proposes a complex methodology with important applications for science, in my opinion, I believe that these results deserves all the attention of the scientific community, even if they are validated only on numerical simulations for the moment;
- Also, have been introduced within the revised paper all the result paragraphs. In my opinion, the paper is improved compared to the proposed of initial version.